# Investigation of the Information Possibilities of the Parameters of Vibroacoustic Signals Accompanying the Processing of Materials by Concentrated Energy Flows

**DOI:** 10.3390/s23020750

**Published:** 2023-01-09

**Authors:** Sergey N. Grigoriev, Mikhail P. Kozochkin, Artur N. Porvatov, Sergey V. Fedorov, Alexander P. Malakhinsky, Yury A. Melnik

**Affiliations:** Department of High-Efficiency Processing Technologies, Moscow State University of Technology “STANKIN”, Vadkovskiy per. 3A, 127055 Moscow, Russia

**Keywords:** laser processing, electrical discharge machining, electron beam machining, vibroacoustic diagnostics, monitoring, spectral analysis, power density, diagnostic parameter

## Abstract

Creating systems for monitoring technology processes based on concentrated energy flows is an urgent and challenging task for automated production. Similar processes accompany such processing technologies: intensive thermal energy transfer to the substance, heating, development of the melting and evaporation or sublimation, ionization, and expansion of the released substance. It is accompanied by structural and phase rearrangements, local changes in volumes, chemical reactions that cause perturbations of the elastic medium, and the propagation of longitudinal and transverse waves in a wide frequency range. Vibrational energy propagates through the machine’s elastic system, making it possible to register vibrations on surfaces remotely. Vibration parameters can be used in monitoring systems to prevent negative phenomena during processing and to be a tool for understanding the processes’ kinetics. In some cases, it is the only source of information about the progress in the processing zone.

## 1. Introduction

Industry development requires the widespread introduction of new efficient technologies based on modern science and technology’s achievements in production. It includes technologies based on the impact on the product surface by concentrated thermal energy flows with a high power density. It is the impact of plasma jets, laser radiation, and electron or ion flows. Accordingly, such technologies are called electron-ion-quantum technologies or radiation-beam technologies (RBT). Such technologies have their specifics, which differ from traditional materials’ surface machining. It allows them to displace traditional technologies for modifying surface layers due to higher efficiency, environmental friendliness, and the uniqueness of the obtained results. RBTs are less energy intensive since only a relatively thin surface layer is processed.

They include laser, electron beam technologies, and electrical discharge machining. All these three directions have their characteristics and application areas, but their unifying feature is the method of influencing the material with a concentrated energy flow (CEF) with a high power density. It implies the commonality of the effects on materials and the properties of diagnostic signals accompanying processing. It is crucial because high-performance CNC equipment has been created for their implementation. Such equipment can be part of flexible production sites that can operate for a certain period without the participation of operators. It is necessary to have operational monitoring systems that control the processing course for these purposes to correct modes and prevent negative situations. Only the control of electrical parameters and vibroacoustic (VA) diagnostics methods can be used from the known monitoring tools for RBT processes. The control of electrical parameters made it possible to create electrical discharge machines with adaptive control. However, the experience of its automatic operation shows that short circuits and wire tool electrode breakage occur, which requires operator intervention. The utilization factor of the discharge current pulses often remains below the optimum. The issue of monitoring is complicated by the use of vacuum chambers in the electron-beam processing of materials that significantly complicate the acquisition of diagnostic information. It is due to the presence of powerful electromagnetic interference at the time of the electron gun discharge and the difficulties of placing the monitoring tools inside the vacuum chamber. The situation is simpler in laser processing than in electrical discharge and electron beam machining since such processing is not associated with the immersion of the workpiece in a working fluid or a deep vacuum. However, the relevance of monitoring remains, making it possible to trace the working process’s kinetics, study the occurring phenomena’s features, and adjust the modes.

Thus, the development of tools that allow studying the kinetics of technologies using thermal energy flows is relevant both for expanding the production capabilities of these technologies and for the in-depth study of physical phenomena that determine the quality and performance of the corresponding equipment. The paper presents the results of studies of the relationship between the parameters of VA signals accompanying RBT processes and the technologies’ features and assesses the possibilities of using VA signals for real-time monitoring.

## 2. Physical Phenomena, the Kinetics of Which Is Studied by Controlling Acoustic Signals

### 2.1. Directions of Research Carried out Using the Control of Elastic Waves

Acoustic emission (AE), in a broad sense, is the radiation of elastic waves that occur in a rearrangement of the internal structure of the substance. AE occurs during the plastic deformation of solid materials [1,2,3,4,5], the development of defects in them [6,7,8], phase transformations associated with a change in the crystal lattice [9,10,11,12,13], the formation of particles of a new phase in supersaturated solutions, and when the boundaries of magnetic domains shift during crystallization and melting of a substance [14,15,16,17], and it is widely used as a method of nondestructive testing in defectoscopy [18,19,20,21,22]. Traditionally, an element of an elastic medium experiencing a change in the stress condition is considered a source of acoustic waves propagating through an elastic system.

The frequency range of registration of acoustic waves is quite broad. The vibrations generated by the waves are recorded in relatively low frequencies (up to 20 kHz) and the ultrasonic range above 20 kHz. In the first case, they are sometimes called vibroacoustic signals, and, in studies of the higher frequency range, they are sometimes called acoustic emissions. One of the mentioned terms is used when using vibration signals covering both ranges. There are accelerometers that register VA signals in ranges up to 100 kHz. In most cases, this range is sufficient to solve the problems of nondestructive testing of technology equipment and the working process condition monitoring, including wear and breakage of cutting tools [23,24,25]. The tasks of the manufacturing industry forced researchers to delve into the analysis of VA signals during friction contact [26,27] and cutting [28,29,30].

### 2.2. The Use of VA Signals to Monitor the Mechanical Machining of Materials

Using the capabilities of VA signals for real-time monitoring of the condition of various tools during processing is especially important in automated production when the equipment operates for a long time without operator control. The use of VA signal parameters is attractive because the primary converter (accelerometer or AE sensor) can be installed at a distance from the processing zone without reducing the versatility of the equipment and interfering with the design of the units. In this regard, accelerometers are much more attractive than AE sensors since they can receive helpful information from the cutting zone at a considerable distance in the frequency range of up to 50 kHz if movable and loosely tightened joints do not separate the cutting zone and accelerometer. The appearance of such joints reduces the signal transmission coefficient, especially at high frequencies. It allows the accelerometer installation on lathes to be conducted from the turret side and, on milling machines from the workpiece side, for example, on the working table. On cylindrical grinding machines, where both the grinding wheel and the workpiece rotate, the accelerometer can be installed on the headstock body of the workpiece: the workpiece rotation velocity is low, although the VA signal weakens when elastic waves pass through the spindle supports; it largely retains its information content. Figure 1 shows the octave spectra for a CNC cylindrical grinder, where the accelerometer was mounted on the headstock. Graphs (1) are given for the period of the work circle after correction, and graphs (2) are for the work circle requiring it. It can be seen that when the wheel is clogged, the effective (root mean squared, RMS) amplitude begins to increase in the low-frequency region and decreases at high frequencies. In this case, standard octave bands were chosen. Nevertheless, even such a simplified representation of the VA signal spectrum may establish that the amplitude decreases in the high-frequency region and increases in the low-frequency region as the grinding time increases in both presented examples. In practice, there are cases where, with an increase in the clogging of the wheel, the VA signal amplitudes increase in the entire frequency range under consideration, but the increase in the low-frequency region is more intense at the same time.

A similar picture of the change in the VA signal spectrum with increasing wear is also observed in the mechanical machining of materials but in the absence of intense self-oscillations [30]. Figure 2 shows examples of the VA signal spectra from the accelerometer accompanying the turning of a roll made of steel 45 (ASTM 1045) with a depth of 0.5 mm, feed velocity of 0.15 mm/rev, and a cutting velocity of 47 m/min. The spectra illustrate two extreme situations: cutting with a sharp insert and an insert with extreme wear, which caused the most significant deformations of the surface layer. On the inset in the corner (Figure 2) the change in the *K_f_* parameter for five inserts with different wear of the cutting edge is shown. The surface layer deformation intensity is shown horizontally as a percentage of the maximum value. The ratio of RMS amplitudes in the low-frequency (0.7–1.5 kHz) and high-frequency (4–15 kHz) ranges was taken as the *K_f_* parameter.

The growth of the *K_f_* parameter indicates a change in the ratio of amplitudes in the low and high frequencies in the VA signal spectrum. At the same time, the VA signal amplitudes can increase over the entire range, but the spectrum should be deformed. An increase in the proportion of viscous fracture relative to a brittle one explains the *K_f_* parameter increase.

The work of the cutting edge is similar to a punch, where the greatest pressure region with a decrease in diameter is localized. The decrease in plastic deformations in the stress concentrator presence indicates that the concentrator presence leads to embrittlement of the fractured material. The cutting edge can be represented as a cylindrical surface with a radius *r*, which is pressed into the shear plane of the chip elements. Theoretical calculations [31,32] show that with the same force action, the maximum stresses *σ_max_* on the cutting edge with a cylindrical surface will increase approximately following the expression (1) in this case:*σ_max_* ~ *r*^−0.5^.(1)

This suggests that with an increase in *r* (with an increase in edge wear), the stress concentration coefficient will decrease, and the force action power density on the separated material layer will decrease. In this situation, the chip formation will shift toward viscous fracture and viscous crack formation. With a decrease in *r*, more brittle cracks will appear during the material destruction with an increase in power density and the stress concentration coefficient. The propagation velocity of brittle cracks is commensurate with the velocity of transverse waves in the material, while the viscous cracks’ velocity may not exceed the material deformation velocity. The crack propagation velocity determines the duration of elastic pulses during crack propagation. The shorter the deformation pulses, the greater the amplitude of the VA signals at high frequencies [30]. Slow cracks accompanying viscous fracture form pulses of relatively long duration, increasing the amplitudes at low frequencies. Figure 2 shows that the amplitudes for frequencies above 5 kHz decreased several times at the extreme wear of the cutting edge in the spectrum of the VA signal. At the same time, the amplitude increased by 20–25% at frequencies up to 2 kHz. The viscous fracture growth is accompanied by an increase in plastic deformations of the workpiece surface layer to a depth of 100 µm or more, which is shown in the inset of Figure 2. The monitoring practice of wear of the cutting edge shows that its shape can only be considered cylindrical in the first approximation, but it is evident that the increase in wear causes an increase in the deformation of the machined surface and the separated chip itself.

### 2.3. Results of the Study of VA Signals during Processing by Concentrated Energy Flows

Processing with concentrated energy flows, which includes laser processing, is considered a noncontact technology. However, an energy flow with a specific power density is also supplied to the machined surface. In laser processing, the power density can be estimated from the input power ratio to the focal spot area. There are few publications in the technical literature devoted to the current evaluation of laser processing performance using various monitoring methods [33,34,35,36,37]. However, assessing the current performance of laser processing using monitoring remains relevant [36,37]. In this regard, we can note the works devoted to studying acoustic signals accompanying laser processing [33,34,35,38]. Figure 3 shows a photo of a laser beam trace when aluminum alloy is being processed and the dependence of the trace depth on the surface to the laser power as a percentage of the maximum power.

Figure 3 shows that the depth of the trace left by the laser changes almost linearly with increasing power. Since the focal spot diameter did not change with a power change, we can say that the depth of the trace increases linearly with a power density increase. Linearity is broken when the laser power is less than 10%. In these cases, the incoming power is enough to heat the workpiece surface, but this power is no longer enough for evaporation and ejection of metal from the trace. A fairer assessment of performance in the form of the removed material volume per unit of time or for the same number of pulses is complicated by the difficulties in determining the released material volume. We have to use approximate calculations due to the trace complex shape. Such experiments were carried out in [34,36]. This allowed us to assert that the volume of the released material increases monotonically with an increase in the power density of the laser radiation, even in this case. It is possible to obtain the dependence of the amplitude of the VA signal parameters on performance by making simultaneous VA signal recordings in laser processing. The analysis of the laser beam traces allowed us to estimate the relationship of the VA signal amplitude with the laser processing performance in the form of power dependence [34,38]:A_16_ = C·M^β^,(2)
where A_16_ is the VA signal’s effective amplitude in the octave band of 16 kHz, M is the volume of released material in mm^3^, C is a constant depending on the selected frequency range for the VA signal and the transmission coefficient of the accelerometer and the entire observation channel, and β is the empirically determined exponent. For example, the indicator was in the range of 1.14–1.16 when processing AISI 410 stainless steel. These values will differ if we use a different frequency range and process different materials. Nevertheless, the monotony of such dependence is important for practical purposes.

However, there are more complex problems associated with laser processing in addition to monitoring the current performance. For example, metal removal can be carried out in the substance evaporation mode. Such conditions provide a higher surface quality. However, the evaporation mode requires excessive energy consumption and may not be optimal. Maintaining a balanced ratio between the melted and evaporated substance is an urgent and difficult task. Another problem may be the change in the position of the focal plane relative to the machined surface. Deviations can be associated with the complexity of the workpiece geometry and the errors in its fastening. The amplitude of the VA signal monitoring in one frequency range is no longer enough to solve this problem by monitoring the VA signal. Figure 4 shows the VA signal spectra in two frequency ranges when processing AISI 410 stainless steel with different laser radiation power.

Figure 4 shows that an increase in the laser radiation power by 1.5 times led to a VA signal amplitude increase in all the presented frequency ranges. However, the increase is noticeably more significant in the high-frequency region. Figure 5 shows the change in the RMS amplitude of the VA signal in the octave bands of 8 and 16 kHz with an increase in the power density q_s_, which is proportional to the laser power in this case. The same figure shows the change in the *K_f_* parameter, which decreases with an increase in q_s_ due to the outstripping growth of VA signal amplitude in the octave band of 16 kHz. The upper part of Figure 5 conditionally shows the scale of change in the heating temperature of the substance with an increase in q_s_. As the power density increases, the melting temperature T_mel_, the evaporation temperature T_evap_, and the ionization temperature T_ion_ of the substance are reached.

If we compare Figure 2 and Figure 5, a direct analogy arises, determined by the dependence of the *K_f_* parameter on the power density of the energy impact on the material. If an increase in power density leads to the formation of fast brittle cracks during mechanical machining (cutting), then an increase in q_s_ leads to the formation of short pulses accompanying the evaporation and ionization processes during laser processing.

Having data on the *K_f_* parameter change in situations where the power density is difficult to estimate, it is possible to monitor the displacement of the laser machining process toward evaporation or melting. Figure 6 shows the spectra of the VA signal accompanying the laser melting of an AISI 410 metal powder layer with different laser beam velocities. The spectra for better visualization are presented separately for the low and high frequencies. It can be seen that an increase in the laser beam velocity leads to an increase in the amplitude of the VA signal in almost the entire frequency range, but the increase in the high-frequency region is more noticeable, especially in the 10–20 kHz range.

Figure 7a shows the change in the effective amplitudes of the VA signal with a change in the laser beam velocity. Graphs are given for amplitudes in the low and high frequencies, and the changes in the *K_f_* ratio are shown. Figure 7b shows photographs of traces obtained by powder melting at different velocities. When the beam moves with a low velocity of 10 mm/s, a large amount of powder is fused, the trace is wide, and the powder consumption is significant. Increasing the beam velocity to 25 mm/s made it possible to reduce the trace width while maintaining uniformity. With an increase in velocity up to 50 mm/s, the trace became thinner, but its uniformity was noticeably disrupted. A decrease in the *K_f_* parameter with an increase in the laser beam velocity indicates that the proportion of the evaporated substance increases with an increase in velocity. At the same time, the proportion of the melt decreases. It leads to a violation of the uniformity of the trace and affects the quality of the resulting product.

Another aspect of using the *K_f_* parameter may be monitoring the position of the focal plane. Changes in the focal plane position relative to the machined surface lead to changes in the focal spot diameter, an increase in its area, and an increase in the *K_f_* parameter. If the permissible deviation is exceeded, there is a reason to adjust the position of the focal plane [35].

Electron beam doping occurs in vacuum chambers and also uses the effect of a concentrated energy flow transmitted by an electron beam with a duration of 5 µm on the surface layer of the target (workpiece). Process monitoring in vacuum chambers is complicated by the impossibility of primary converters being placed in a chamber with a high vacuum and a significant level of electromagnetic interference occurring at the electron beam discharge. Vacuum coating methods are now widely used. These methods can be the most rational solution to increase critical parts’ wear resistance and durability. At the same time, the control of surface doping is complicated by a sufficiently large number of input factors with a random component. First, it refers to the thickness of the surface layer, into which is introduced the energy sufficient for its heat processing (melting and partial evaporation). These factors also include the magnitude of the charging voltage, on which the specific energy of the beam depends; the law of the distribution of beam energy over the irradiated surface; the thickness of the film with doping components deposed using magnetron sputtering; a number of processing cycles, etc. Their optimization is carried out based on metallographic, X-ray, and spectral studies [39] conducted after processing.

However, the electron beam parameters’ instability and the process of its interaction with the material to be processed and the random distribution of the beam energy over the irradiated surface lead to significant changes in the results that occur spontaneously, regardless of the control system. It violates the strict surface doping repeatability, especially since no observable parameter reflects its kinetics [40].

It is not easy to find information in the literature on methods for monitoring processes that occur in vacuum chambers during electron beam doping. The use of VA signals for monitoring processes in vacuum chambers was first proposed in [35,39,41]. It was proposed to use a waveguide connecting the processing zone and exiting the chamber through a vacuum inlet to register the VA signals accompanying the processes in the material surface layer after the electron pulse is supplied [35]. A copper wire with an area of 2.3 mm^2^ in cross-section was used as a waveguide. The experiments were carried out on a RHYTHM-SP unit with an electron beam source combined with two magnetron sputtering systems on one vacuum chamber [40,42]. When an electron beam pulses, the surface layer’s substance is subjected to thermal shock. This creates a local volumetric expansion that generates elastic waves in the workpiece and the technological system elements in contact with the workpiece, including the waveguide. The accelerometer was installed at another waveguide end at a distance of up to 2 m from the unit.

The reasons for the occurrence of VA signals during electron beam doping are basically the same as during laser processing. In addition to changes in the local volumes of the material, the sources of elastic waves are changes in the microstructure of the surface layer and pulses of the reactive vapor pressure of the superheated substance. When volumes, internal stresses, and other characteristics change abruptly, it is crucial to observe changes in amplitude over time. However, it is necessary first to use the signal spectrum analysis to select the most informative frequency ranges. These ranges are chosen based on experiments. The signal-to-noise ratio in these ranges should be satisfactory, and the amplitudes should be sensitive to changes in the processing area. Choosing such frequency ranges in which the amplitude changes will not repeat each other, differing only in scale, is important. Therefore, one range was chosen in relatively low frequencies, and the second was in the region of high frequencies. Figure 8 and Figure 9 show the results of experiments with irradiation of an AISI 439 (8Cr17Ti) alloy plate surface as an example of in-time records of the VA signal that displays the kinetics of processes that occurred in a vacuum chamber in electron beam doping. The steel insert was previously subjected to low-temperature nitriding before surface doping. The experiments were carried out with a nitrided insert at the first stage and after Nb70Hf22Ti8 alloy coating deposition to the nitrided surface by magnetron sputtering at the second stage [35].

Figure 8b shows that the maximum of the VA signal in the range of 26–33 kHz is ahead of the maximum of the low-frequency component by 13–14 ms. It suggests that the output of the evaporated material occurs before the formation of waves of thermoelastic stresses, chemical reactions, and crystallization.

The low-frequency component is still present on the recording for a long time after the high-frequency signal fades. Since the VA signal parameters change over time in different ways, the value of the *K_f_* parameter will also change over time. The high-frequency VA signal indicates the change in the proportion of the evaporated substance. Therefore, the *K_f_* parameter value was estimated at the moment t*, when the amplitude A_2_ reached a maximum. In Figure 8, the ratio *K_f_* (t*) = 2.5. The value is compared with the results obtained after the Nb70Hf22Ti8 alloy doping film deposition on the steel surface. Figure 9 shows VA signal recordings in two frequency ranges and the RMS envelope of these recordings under conditions similar to Figure 8.

When comparing Figure 8 and Figure 9, it can be seen that the VA component of the A_2_ signal quadrupled. Considering the *K_f_* parameter at the moment t*, it became 0.5 for the VA signal in Figure 9 instead of 2.5 in Figure 8. This is due to the intense evaporation of the Nb70Hf22Ti8 alloy coating. After irradiation of such a sample with an electron beam, the coating and substrate materials are mixed, and an exothermic chemical reaction is triggered with the formation of nitride phases based on niobium and hafnium. Attention is drawn to the fact that the VA signal fades over time. Figure 9 shows that the attenuation goes with wave modulation. The moment can be interpreted as follows: at the initial moment, there is an intense heating of the material, accompanied by evaporation, which triggers chemical reactions and phase transitions in the material. It can be assumed that such behavior of the VA signal is associated with a martensitic transformation in several stages caused by a chemical reaction of the formation of a nitride phase based on niobium. The stages of martensitic transformation do not occur in one region but sequentially trigger martensitic transformation in several regions through plastic deformation [41,43,44].

Figure 10 compares the spectra under irradiation of an uncoated aluminum insert and an insert coated with Ni77Cr23 film.

Figure 10 shows that the amplitudes of the VA signal upon irradiation of pure Al are many times higher than the amplitudes upon irradiation of an insert coated with Ni77Cr23 film up to a frequency of 14 kHz. The changes are in the high-frequency region. The amplitude increases sharply at high frequencies when irradiating a surface coated with Ni77Cr23 film. At the same time, it is even more sharp at some frequencies when irradiating pure aluminum. After a series of experiments, the condition of the surfaces was studied using the Thixomet image analyzer to estimate the proportion (in %) of the area occupied by intermetallic compounds. The obtained estimates of the process performance were compared with the RMS amplitude values in octave bands with central frequencies of 16 and 32 kHz (Figure 11).

The experiments were carried out under the same conditions. Still, the uncertainty of some factors led to a spread of results relative to the area of the intermetallic coating. The example in Figure 11 shows the possibility of evaluating the qualitative indicators of the operation in real-time. At the same time, it should be considered that too small values of the *K_f_* parameter can occur due to large amplitudes of the VA signal at high frequencies. The situation may indicate excessive evaporation of the applied film. It is necessary to set a lower limit for the *K_f_* parameter to limit evaporation and promptly limit the discharge voltage when it is reached.

Figure 12 shows a VA signal example of another negative phenomenon associated with the appearance of surface cracks 2 s after the electron pulse is applied. The record is shown in two frequency ranges. It can be seen that the pulse associated with the crack formation in the VA signal recording appears only in the low-frequency range. This suggests that the pulse duration is longer than the pulses generated by evaporation during irradiation. Such information makes it possible to promptly reirradiate the damaged surface layer for remelting and to change the irradiation mode.

To assess the relationship of the VA signals’ parameters with the power density of the energy impact with electron beam technology, the discharge voltage of the electron gun was changed during the irradiation of different materials. Figure 13 shows the results of experiments with irradiation of the M20 (MS 221) alloy. The arrow in the graph shows the direction of change in the power density with a discharge voltage increase. It can be seen that the low-frequency component of the VA signal changes little with increasing q_s_, but there is a rapid increase in the range of 30–40 kHz after 18 kV, providing a decrease in the *K_f_* ratio. Therefore, it was necessary to introduce a limit on the discharge voltage at 22 kV.

Thus, when electron beam processing, the behavior features of VA signal parameters noted during laser processing and mechanical machining (cutting) are preserved: an increase in the power density of the energy acting on the material to be processed leads to a deformation of the spectrum of the generated VA signal in the direction of a progressive increase in high-frequency components.

## 3. Comparison of Acoustic Properties of EDM and RBT Processes

Electrical discharge machining technologies refer to technologies that affect the material to be processed by concentrated energy flows, similar to laser and electron beam processing. Thermal energy is transferred here by flows of electrons or ions. From the results obtained earlier for laser and electron beam processing, it can be concluded that VA signals will arise, and they will be the patterns of changes that should correspond to the previously obtained data for laser and electron beam processing during electrical discharge processing.

Electrical discharge machining (EDM) is one of the most high-precision processes: modern machines make it possible to obtain products with an accuracy of up to 1 μm (the positioning step is up to 80–100 nm) [45,46]. Between four and six independent axes significantly expand the scope of the technology, and the use of modern dielectrics makes the process safer for the operator’s health and the environment. EDM has become an indispensable technology in manufacturing high-precision parts from difficult-to-process materials for critical purposes, with complex spatial geometry and inner cooling channels. In modern production, processing diagnostic tools are becoming more widespread. Particularly relevant are the means of online control during machining. In the EDM, the interaction occurs with the workpiece immersion in the dielectric fluid. The processing zone is so small and remote from its surface that it is difficult to use a wide range of diagnostic tools. However, among the technologies that use high-energy flows to impact the material, EDM is distinguished by an adaptation of processing modes based on the analysis of electrical parameters. The problems of further improvement of EDM and the creation of equipment capable of functioning without an operator require research to improve EDM monitoring methods, which are especially relevant for modern industry. Methods and means of VA diagnostics of the EDM will expand knowledge about the physics of the technology and create a set of measures that will improve the quality and reduce EDM defects in manufacturing critical products.

From the mentioned physical phenomena that occurred in EDM (Figure 14), the same phenomena listed above (Section 2.3) during laser processing can be seen. These are surface melting–evaporation/sublimation, destruction/dissociation, wells’ formation, recombination/deposition, cooling, and crystallization. Based on this view, it can be expected that VA phenomena accompanying EDM will have similar properties, such as (1) the monotonic relationship between the amplitude of the VA signals and the machining performance; (2) a change in the ratio of amplitudes in the low-frequency and high-frequency ranges of the accompanying VA signals with a change in the ratio of the volumes of the melt/decomposed substance and the substance in the condition of sublimation/volumetric boiling and evaporation.

One of EDM’s actual problems is assessing the interelectrode gap (IEG) condition. Solutions to these problems with the help of control of electrical parameters have the following disadvantages: the dependence of the results on the modes, area, depth of processing, and flushing modes; the difficulty of obtaining information at high frequencies of discharge pulses with short pauses. These disadvantages are especially evident in processing modern ceramics with relatively low electrical conductivity. When processing such materials, wire tool electrode breakage, low-pulse utilization, insufficient quality of the resulting surfaces, cracks, and chips can occur.

The IEG condition constantly changes due to the imbalance between the newly recombinated/formed erosion products and the ones to be removed. If the inflow of erosion products exceeds the small volume of the IEG, then the electrical conditions (resistance) of the IEG change. Part of the supplied energy is spent on the destruction/dissociation of erosion products, and the power density supplied to the workpiece decreases. This should affect the change in the spectrum of the VA signal accompanying the EDM. Figure 15 shows the VA signals spectra when HG20 (W94K6) alloy workpiece electrical discharge machining. Graphs marked (1) display the spectra from the machining beginning, and graphs marked (2) are before the wire tool electrode breakage.

The spectra of the VA signal (Figure 15) show that before the wire tool electrode breaks, the spectrum changes: the amplitudes at high frequencies decrease significantly, and they increase at low frequencies. Previously, it was explained by a decrease in the power density of the energy impact. In EDM, the drop in power density is associated with an increase in the erosion products’ concentration in the IEG. Part of the energy of the discharge pulse is spent on the destruction and dissociation of particles released into the working fluid, decreasing the power density q_s_. Figure 16 shows the record of the VA signal components in different frequency ranges: 1.4–2.8 kHz and 10–20 kHz. The record is made from the beginning of machining to the wire tool electrode breakage.

Figure 16 shows that the amplitude of the VA signals in two frequency ranges behaves oppositely as electrode breakage approaches: the amplitude decreases at high frequencies and increases at low frequencies. The above experiments show that such a situation corresponds to a drop in the power density q_s_. Figure 16 shows that the change in the amplitude of the VA signals is not monotonous but is accompanied by peaks and drops. The trend toward increasing at low frequencies and dropping at high frequencies is visible. Amplitude fluctuations are associated with instability in the interelectrode gap due to a nonmonotonic increase in the concentration of erosion products. Figure 17 shows the RMS envelopes of the records of Figure 16 and the change in the *K_f_* parameter in time.

The interpretation of the processes shown in Figure 15, Figure 16 and Figure 17: over time, the concentration of erosion products in the IEG increases, and the power density q_s_ decreases. It reduces the proportion of the evaporated/sublimated substance and worsens the release of the recombinated material into the working fluid. As a result, well overlaps are created along the crater’s edges, and a favorable situation for the localization of discharges is created (Figure 18). A decrease in the volume of the evaporated/sublimated substance leads to a decrease in the VA amplitude at high frequencies. The heating of additional volumes promotes the growth of relatively slow thermoelastic pulses, contributing to the VA signal amplitude at low frequencies.

Figure 18a shows a photograph of the trace from the wire tool electrode after the breakage. The overlaps of wells on the surface can be seen. Figure 18b shows a profilogram of the trace, where there is a dominant protrusion that was not present at the initial stages of machining. Figure 18c is a picture of the wire tool electrode on a scanning electron microscope with traces of adhering material. The results of the temperature field calculation by the finite element method are shown in Figure 19a. The temperature fields in the tool and workpiece electrodes are shown during normal processing (discharges are evenly distributed over the surfaces of the electrodes) and during the localization of discharges. Localization of discharges leads to the heating of the electrodes to a considerable depth. At the wire tool electrode, heating covers the entire cross-section of the wire, reducing its modulus of elasticity. As a result, there is a breakage. When processing conductive ceramics, an increase in temperature is accompanied by cracks and chips. An example of processing VOC60 (Al_2_O_3_ + TiC) oxide-carbide ceramics is shown in Figure 20. It can be seen that chips and cracks appeared next to the trace of the wire tool electrode.

Figure 21a shows the spectra of VA signal with a short circuit (SC), which occurred in the HG20 conductive alloy processing. It can be seen that the amplitudes drop several times in a wide frequency range but not to zero. The interference from the operation of other machine mechanisms is preserved. In addition, the heating of the surface leads to a local change in volume and structural and phase rearrangements that create an acoustic background. Figure 21b shows parallel recordings of the signal and current envelopes in the area with short circuits. Two limits, L1 and L2, are shown. When the VA amplitude drops below L1 and the discharge current exceeds the limit of L2, SC can be registered.

The situation becomes more complicated in electrical discharge machining ceramics with relatively low electrical conductivity. When SC occurs, the current may change slightly due to the high resistance of the ceramics. In this case, the machine control system takes a long time to identify the moment of short circuit occurrence, which leads to wire tool electrodes heating and breaking. Figure 22 shows parallel recordings of the maximum and minimum values of the discharge current signals and the VA signal in the high-frequency region during the IN23 (Al_2_O_3_ + TiC) ceramics processing. The arrows on the record show two areas with SC. On the record of the VA signal in the SC section, the extreme signal values approach zero. On the current recording in the SC sections, the distance between the extreme values expands slightly, but without a parallel recording of the VA signal, it is not easy to notice this expansion. For this reason, wire electrode breakages occur due to the difficulty of identifying short circuits.

In die-sinking with a hollow electrode, the occurrence of short circuits does not threaten the tool electrode breakage but threatens to cause cracks and chips in processing ceramics. Die-sinking is often carried out when the working fluid is supplied by irrigation. Failures in the fluid supply or a lack of supplied volumes lead to temperature violations and the appearance of cracks. However, the regularities mentioned above of VA signal parameter changes are preserved even in this type of EDM. Figure 23 shows an example with VA signal recordings in two frequency ranges and a change in the *K_f_* ratio in die-sinking IN23 ceramics for 30 s. The records (Figure 23a) show numerous bursts; some are in both frequency ranges depending on the crack development velocity. Despite the crack formation, the patterns of changes in the components of the VA signal components and the *K_f_* parameter remain (Figure 23b), as for other RBT processes.

Figure 24 shows the spectra of the VA signal in two frequency ranges at the initial stage of processing and after 30 s of IN23 ceramics machining to a depth of 6 mm.

The spectra of the VA signal show that some amplitudes increased by three times by 30 s in the frequency range of 3–12 kHz, and they fell by two or more times in the frequency range of 17–39 kHz. The situation arises due to the erosion products’ accumulation in a blind hole when part of the supplied energy is spent on their destruction. Accordingly, this reduces the power density q_s_ with all the ensuing consequences inherent in all processes when the material is exposed to concentrated energy flows.

In EDM, the value of q_s_ also depends on the size of the discharge channel. It is difficult to observe the parameters of the discharge channels directly, but we can get an idea of their change by controlling q_s_. During laser processing, an increase in laser power while maintaining the size of the focal spot proportionally increases the power density, which is reflected in the parameters of the VA signal (Figure 5). In EDM, an increase in the current amplitude does not have a similar effect. Experiments with die-sinking of low-carbon steel and ceramics confirm this. Table 1 and Table 2 present the VA signal parameters and the processing performance as the depth (L) of the hole obtained in 30 s. All conditions were the same; only the current amplitude was varied. Figure 25 shows the dependences of the RMS amplitude of the VA signal at low (A_1_) and high (A_2_) frequencies on the performance L when processing IN23 ceramics.

It can be seen that the amplitudes of the components of the VA signal monotonically grow with the increase in performance. At the same time, the *K_f_* parameter shown in the right column of Table 1 and Table 2 changes very slightly within the measurement error. Thus, the power density changes little with an increase in the specified current amplitude. From previous studies, it can be concluded that the size of the discharge channel increases. The average value of the current increases, and the amount of released and dissociated material increases, but the proportion of evaporated/sublimated material almost does not change.

The experience of electrical discharge machining ceramics shows that the most important task of monitoring the machining is to control cracking. The variety of ceramics does not allow us to determine the rational processing modes in advance, guaranteeing the absence of cracks and chips. It is impossible to observe the occurrence of cracks during EDM using the control of electrical parameters, and there are no other means of registering cracks in real-time except for the acoustic one.

Figure 26 shows the records of the discharge current and the components of the VA signal in three frequency ranges.

A shortage of working fluid accompanied the processing; therefore, numerous areas with no breakdown and short circuits can be seen on the recordings. VA recordings are accompanied by many pulses resulting from the formation of cracks. At the same time, it can be seen that part of the pulses appears only in the lowest frequency range, and part of the pulses is visible at high frequencies. The higher the crack propagation velocity, the more visible the pulse is at higher frequencies. The VA pulses’ size may indicate the formed cracks’ size.

On the record (Figure 26), there are areas where the VA signal is present only at low frequencies but not at high frequencies. That indicates the absence of evaporation/sublimation or its insufficient intensity. In these areas, the material is only heated but is not released (dissociated and recombined) into the working medium.

Figure 27 shows photographs of a hole in a ceramic workpiece, during the processing of which signals in Figure 26 were recorded.

Rough, uneven edges of the hole with chips are visible, and the walls of the hole with cracks are shown. The picture gives the impression that the separation of the material was not in small portions but in large chips.

## 4. Conclusions

It is known from the technical literature that all CEF technologies are associated with similar processes accompanied by structural and phase rearrangements, abrupt changes in local volumes, and elastic stresses that generate elastic waves in a wide frequency range, propagating through the elastic system of the equipment.Studies of vibration signals for various CEF technologies have shown that the frequency range of up to 50 kHz is quite informative for monitoring. This frequency range is more suitable for the industry than the one above 100 kHz, which is often mentioned in the literature when studying, for example, laser impact on materials. Signals at low frequencies decay more slowly with distance and better overcome the joints of parts. Thus, they are more convenient to use in real conditions.Experiments have shown that the vibration signal amplitude during CEF processing increases with increasing heat flux power density in a wide frequency range, which is consistent with the literature data. Nevertheless, the experiments have shown that the amplitudes at higher frequencies grow faster with increasing power density than at low frequencies. An analysis of the phase transitions of a substance with an increase in the heat flux power density, known from the literature, made it possible to compare the transition of a substance from melting to volumetric boiling and evaporation with a change in the ratio RMS of the vibration signal amplitudes in the low- and high-frequency regions (*K_f_* parameter).An increase in the concentration of erosion products during EDM threatens short circuits and wire tool electrode breakage. Erosion products dissipate incoming energy, reducing power density. An increase in the *K_f_* parameter precedes the moment. The control of the *K_f_* parameter makes it possible to prevent an excessive concentration of erosion products and to carry out the relaxation of the working area timely.Control of vibration parameters can be used in processing the ceramics to register the moments of crack formation and subsequent correction of processing modes. Monitoring vibration signal parameters may be the only tool to investigate various materials’ difficult-to-observe CEF processing kinetics.

## Figures and Tables

**Figure 1 sensors-23-00750-f001:**
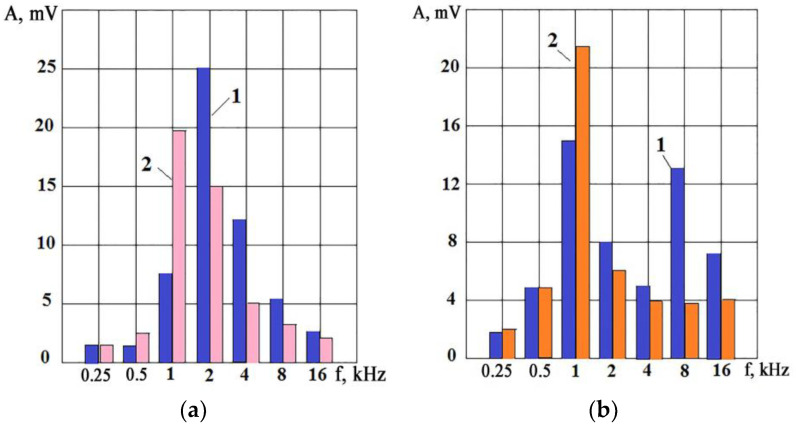
Octave spectra of the VA signal during grinding of steel 45 (ASTM1045) on a CNC circular grinding machine: (**a**) internal in-feed grinding (rotation velocity of the circle = 12,000 rpm, of the spindle = 150 rpm, t = 0.1 mm, circle diameter = 50 mm); (**b**) external longitudinal grinding (rotation velocity of the circle = 1200 rpm, of the spindle = 100 rpm, t = 0.2 mm, circle diameter = 750 mm). (**1**)—circle after editing; (**2**)—worn circle.

**Figure 2 sensors-23-00750-f002:**
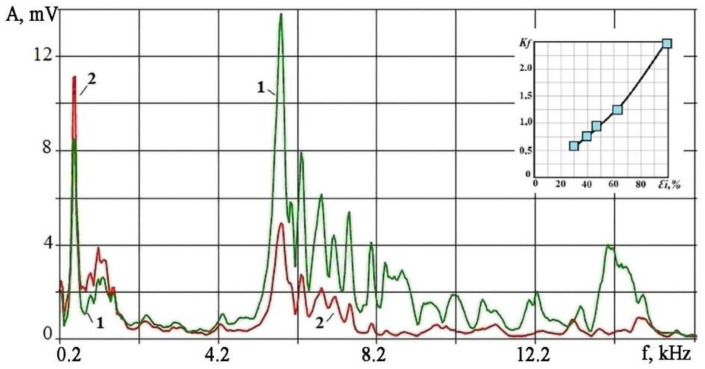
VA signal spectra during turning with sharp (spectrum 1) and worn (spectrum 2) inserts; the inset shows the change in the values of the *K_f_* parameter with an increase in the effect of wear on the surface layer deformation (horizontally shown is the surface layer deformation at a depth of 0.1 mm in percent).

**Figure 3 sensors-23-00750-f003:**
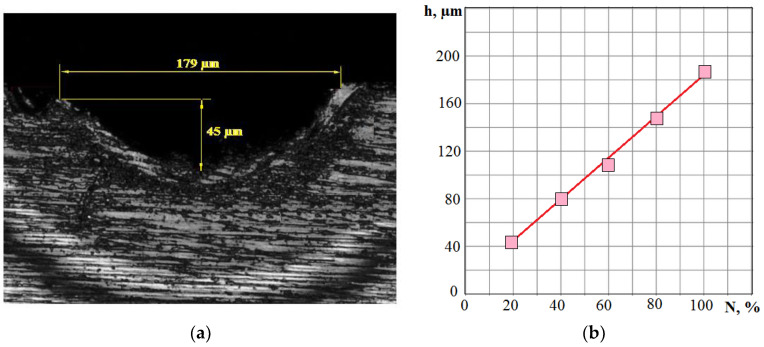
A photograph of the cross-section of the trace on the aluminum alloy surface after the laser beam discharge with 20% of the maximum power (**a**); the trace depth dependence on the laser power (**b**).

**Figure 4 sensors-23-00750-f004:**
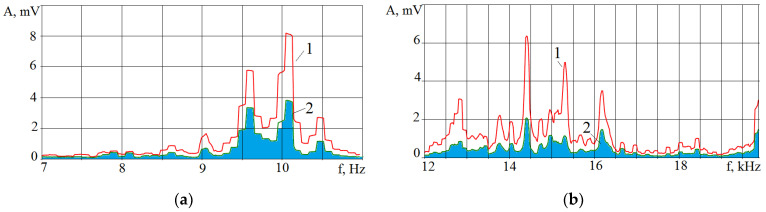
Comparison of the spectra of VA radiation during laser processing with a power of 60% (graphs 1) and 40% (graphs 2): (**a**) low-frequency range; (**b**) high-frequency range.

**Figure 5 sensors-23-00750-f005:**
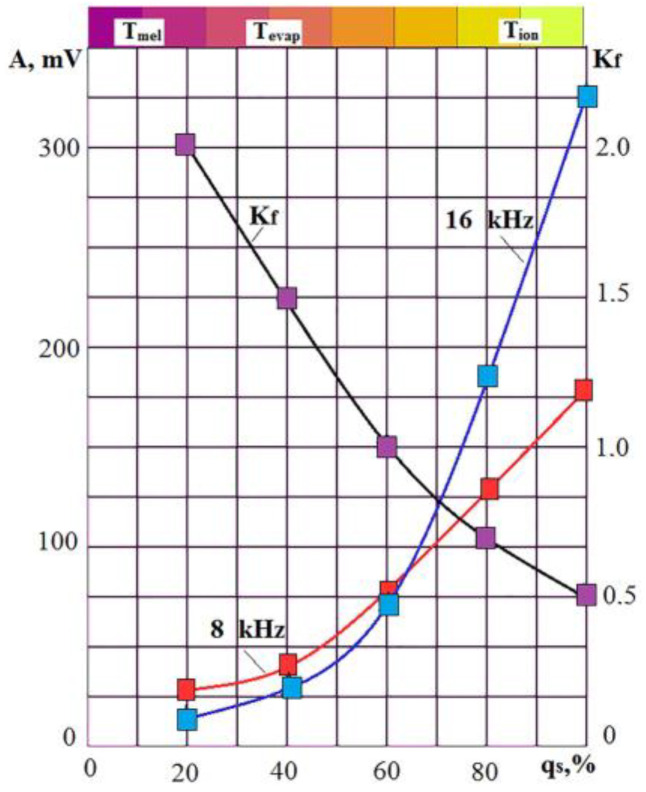
Change in the effective amplitude of the VA signal in the octave bands of 8 and 16 kHz and their *K_f_* ratio when the power density q_s_ of laser pulses changes.

**Figure 6 sensors-23-00750-f006:**
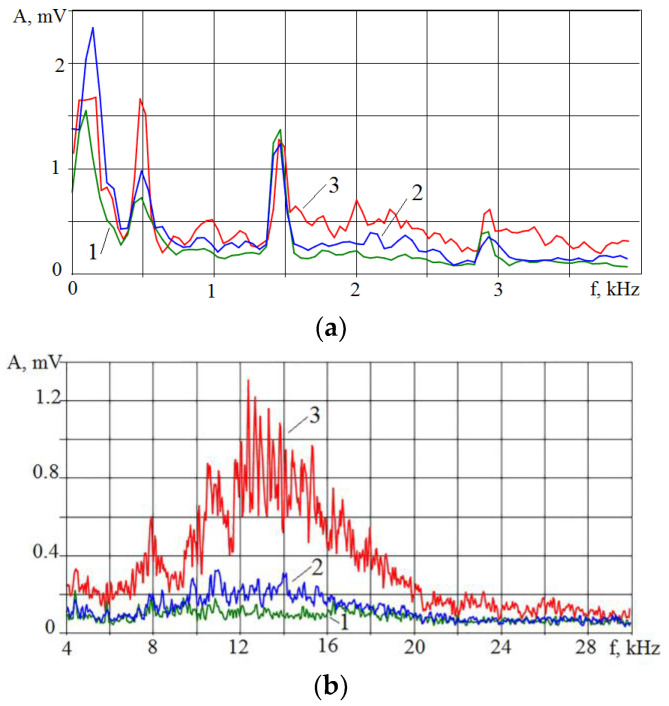
Spectra of the VA signal in the low-frequency (**a**) and high-frequency (**b**) ranges during melting of the AISI 410 powder layer with the different laser beam velocities: (**1**) 10 mm/s; (**2**) 25 mm/s; (**3**) 50 mm/s.

**Figure 7 sensors-23-00750-f007:**
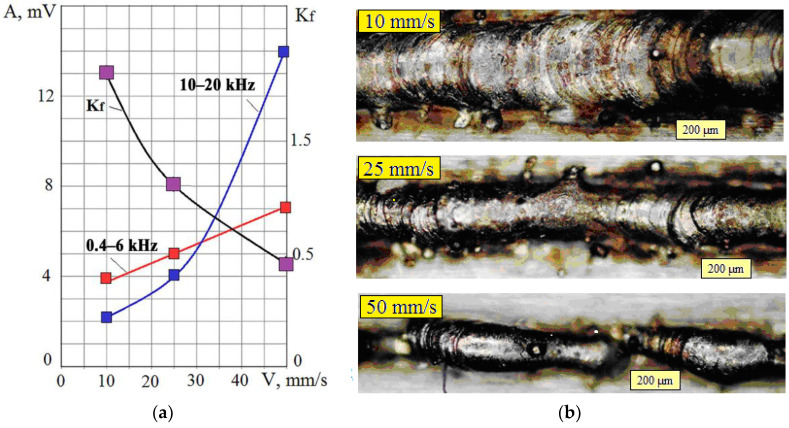
Change of RMS amplitude of the VA signal in the frequency ranges 0.4–6 kHz and 10–20 kHz and the *K_f_* ratio when changing the laser beam velocity (V) (**a**); photos of traces with the different laser beam velocity (**b**).

**Figure 8 sensors-23-00750-f008:**
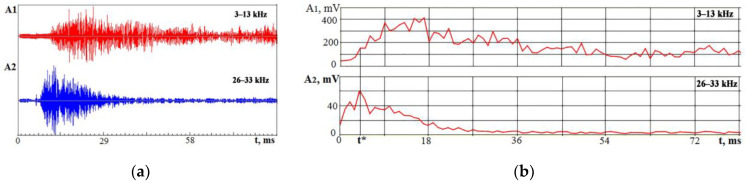
VA signal amplitude changes in time in two frequency ranges under electron beam action on a steel plate at a charging voltage of 22 kV: (**a**) VA signal recordings; (**b**) RMS amplitude of VA signals changes in time. t*—the moment, when the amplitude A_2_ reached a maximum.

**Figure 9 sensors-23-00750-f009:**
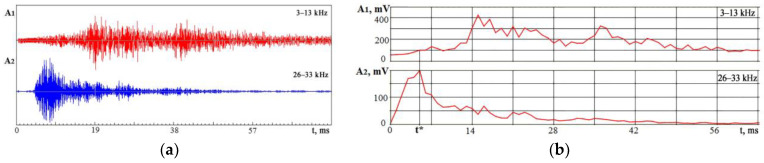
VA signal amplitude changes in time in two frequency ranges during electron beam action on a steel plate with a layer of Nb70Hf22Ti8 film at a charging voltage of 22 kV: (**a**) time records of VA signal components; (**b**) RMS amplitude changes in time for two frequency bands. t*—the moment, when the amplitude A_2_ reached a maximum.

**Figure 10 sensors-23-00750-f010:**
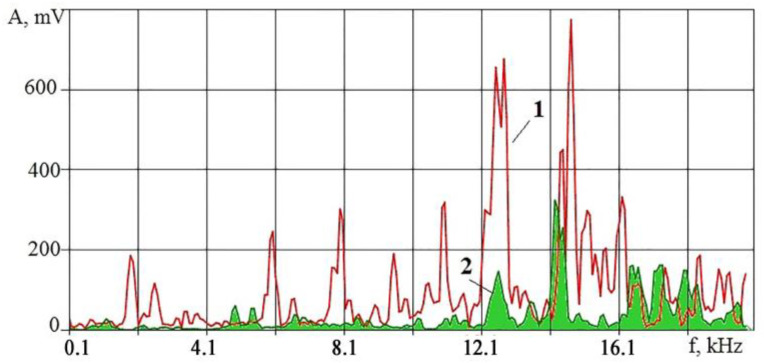
Comparison of the VA signals spectra in the range of 0.1–20 kHz when irradiating an aluminum insert (spectrum 1) and an insert coated with Ni77Cr23 film (spectrum 2).

**Figure 11 sensors-23-00750-f011:**
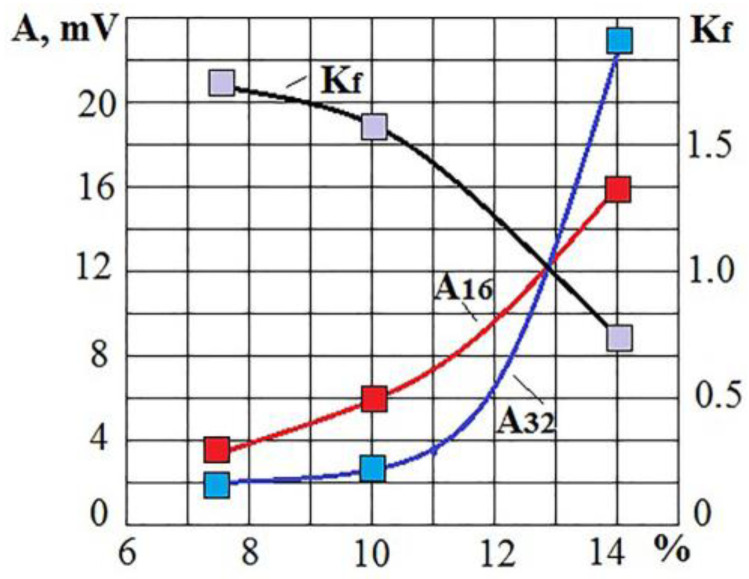
Change in RMS amplitude in octaves of 16 and 32 kHz and the *K_f_* parameter depending on the percentage of the area covered by intermetallic compounds.

**Figure 12 sensors-23-00750-f012:**
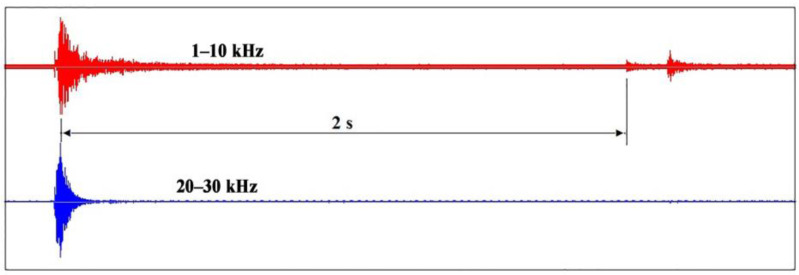
Registration of the moment of crack formation through recording of the VA signal in time in the frequency ranges of 1–10 and 20–30 kHz during irradiation of the M20 alloy with a discharge voltage of 20 kV.

**Figure 13 sensors-23-00750-f013:**
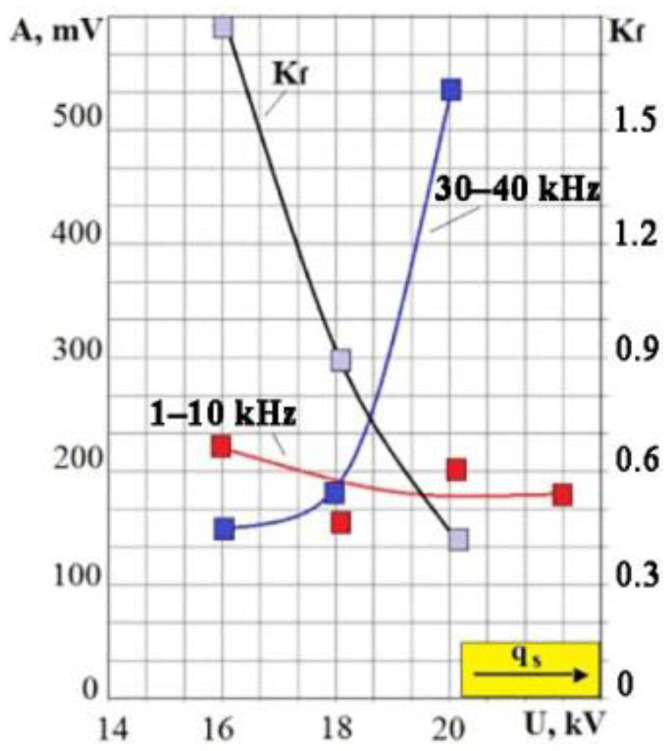
Dependence of the VA signal parameters on the discharge voltage (power density) when irradiating the M20 alloy.

**Figure 14 sensors-23-00750-f014:**
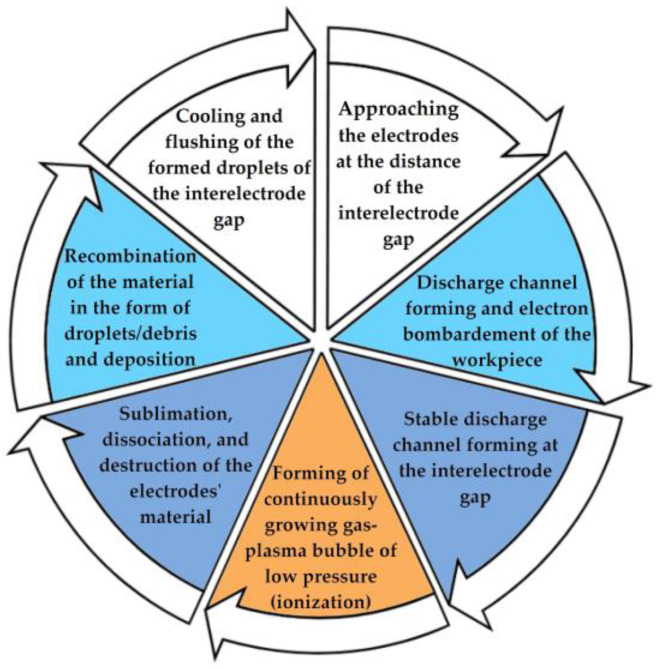
Physical phenomena accompany electrical discharge machining.

**Figure 15 sensors-23-00750-f015:**
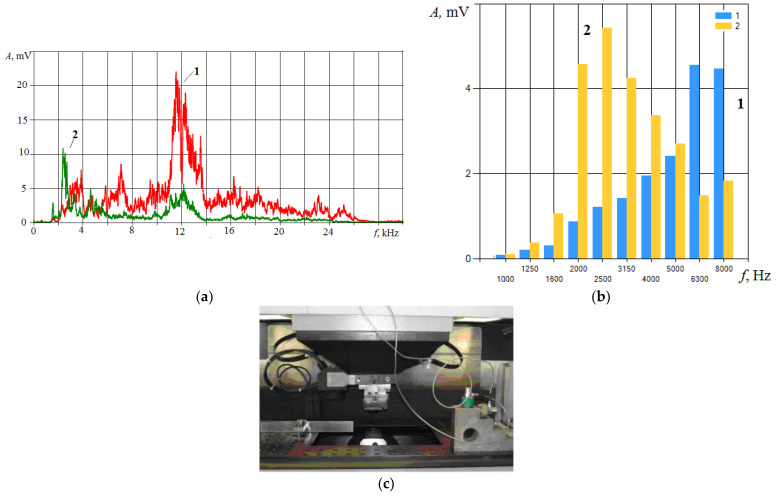
Examples of VA signal spectra during cutting of a workpiece of HG20 alloy: (**a**) amplitude spectrum; (**b**) 1/3 of octave spectrum; where for (**a**) and (**b**) (**1**) is at the machining beginning and (**2**) is at 1 s before the wire tool electrode breakage; (**c**) the working area of the wire electrical discharge machine with installed accelerometers.

**Figure 16 sensors-23-00750-f016:**
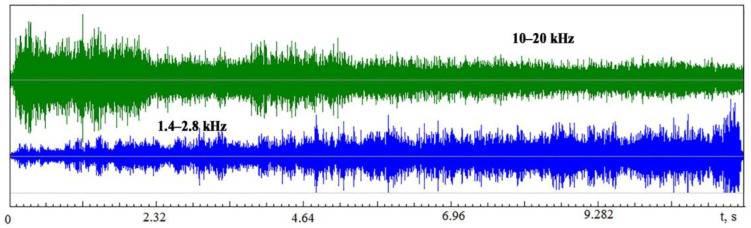
Records of VA signal in time in two frequency ranges from the machining start to the electrode breakage.

**Figure 17 sensors-23-00750-f017:**
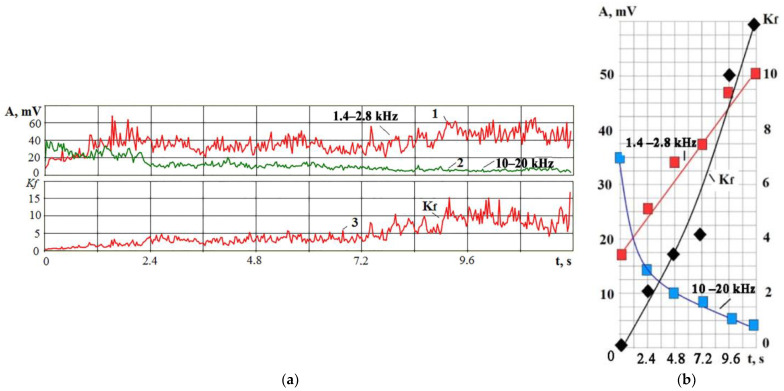
Change of RMS amplitude of VA signals in two frequency ranges (curves **1** and **2**) and *K_f_* parameter (curve **3**) in time: (**a**) continuous recording of signals in time until the wire tool electrode breakage; (**b**) the result of RMS averaging of the record divided into 6 intervals.

**Figure 18 sensors-23-00750-f018:**
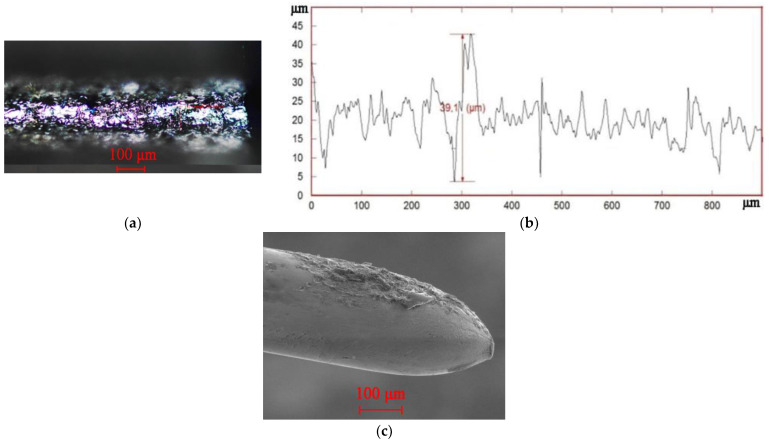
Surface characteristics after wire tool electrode breakage: (**a**) trace on the surface after breakage; (**b**) profilogram of the traces; (**c**) scanning electron microscopy of the wire tool electrode after breakage.

**Figure 19 sensors-23-00750-f019:**
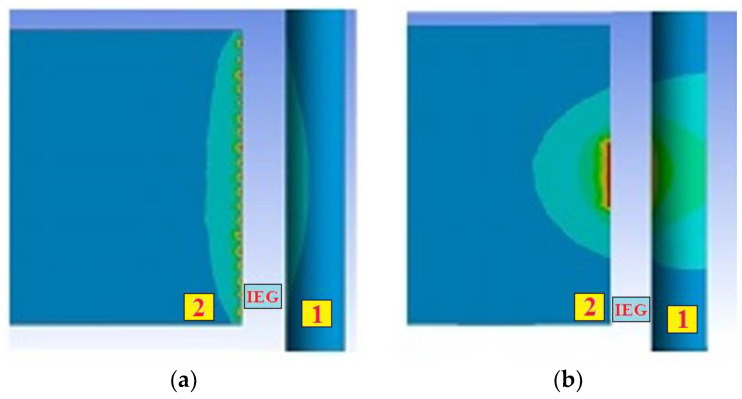
Examples of calculating temperature fields by the finite element method during the normal course of the EDM (**a**) and during the localization of discharges (**b**), where (**1**) is the tool electrode; (**2**) is the workpiece electrode.

**Figure 20 sensors-23-00750-f020:**
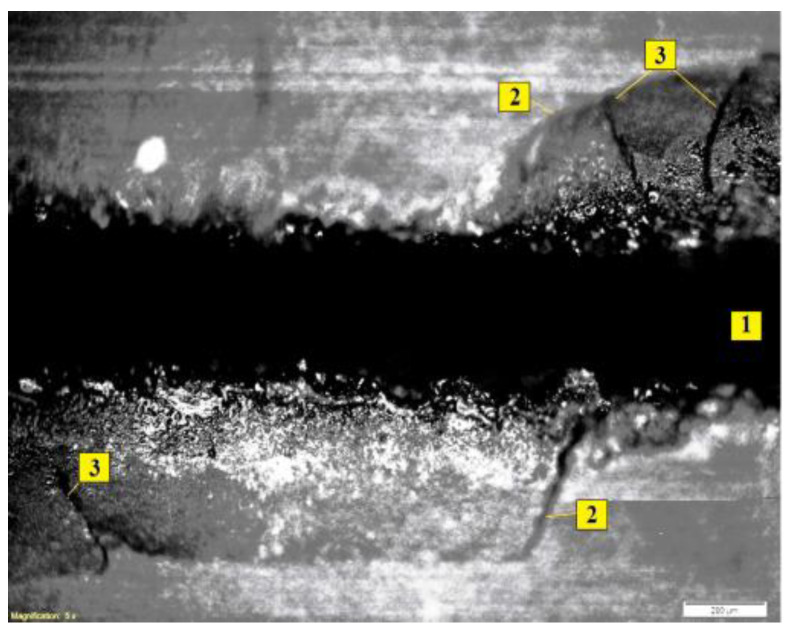
Chips and cracks in the EDM of oxide ceramics: (**1**) is the trace of the tool electrode; (**2**) is chips; (**3**) is cracks.

**Figure 21 sensors-23-00750-f021:**
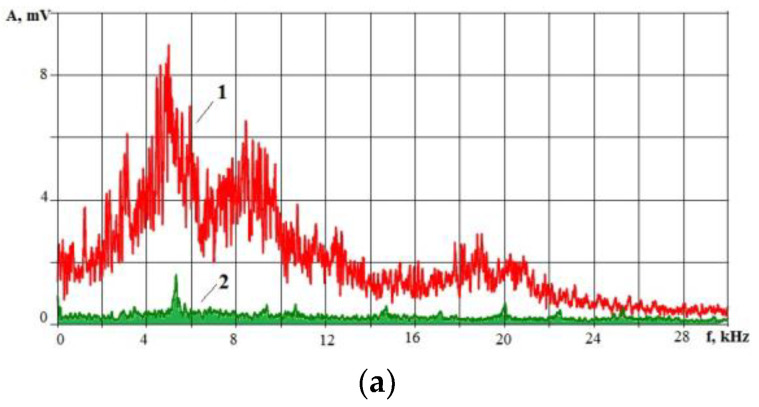
Changes in VA signal and current during short circuit: (**a**) VA signal spectrum during machining (**1**) and short circuit (**2**); (**b**) changes in time RMS of the signal amplitude VA (Av) and current (Ai) during short circuit.

**Figure 22 sensors-23-00750-f022:**
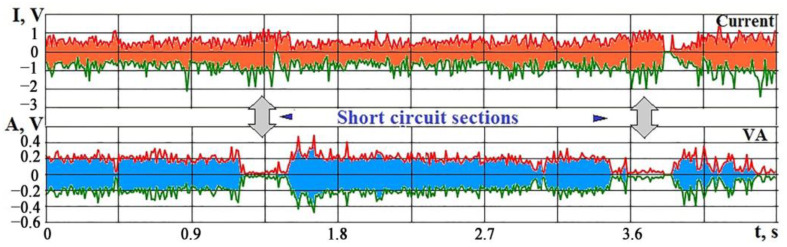
An example of parallel time recording of maximum and minimum current values and VA signal in electrical discharge machining of IN23 ceramics with short circuit sections.

**Figure 23 sensors-23-00750-f023:**
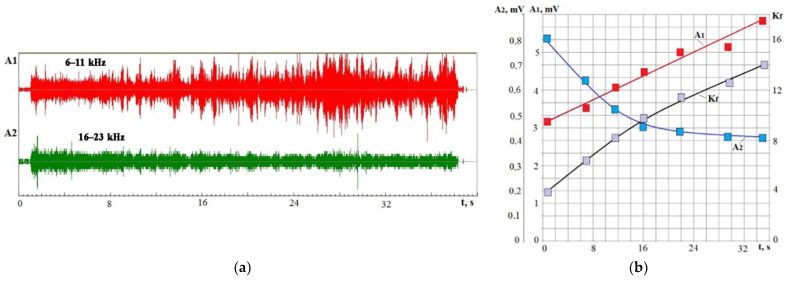
Time recording of VA signals in different frequency ranges in IN23 ceramics die-piercing: (**a**) time recording of signals with amplitudes A_1_ and A_2_; (**b**) change in time of the RMS amplitudes A_1_ and A_2_ and parameter *K_f_*, obtained by dividing the record into 7 intervals.

**Figure 24 sensors-23-00750-f024:**
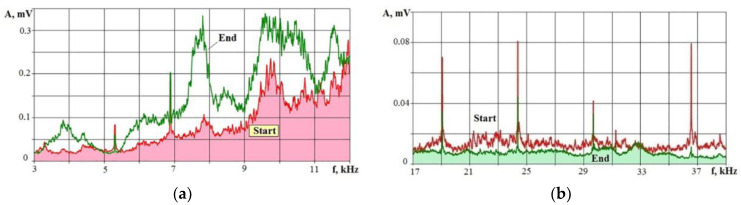
VA signal spectra for two frequency ranges for the initial and final stages of IN23 ceramics processing: (**a**) low frequency range; (**b**) high frequency range.

**Figure 25 sensors-23-00750-f025:**
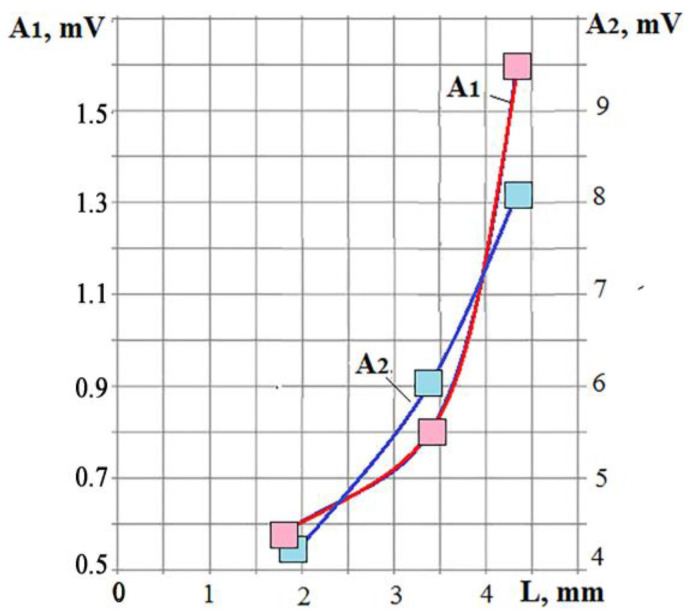
The relationship of the parameters of the VA signals with the die-sinking performance L, where A_1_ is of 1.5–3 kHz; A_2_ is of 5–11 kHz.

**Figure 26 sensors-23-00750-f026:**
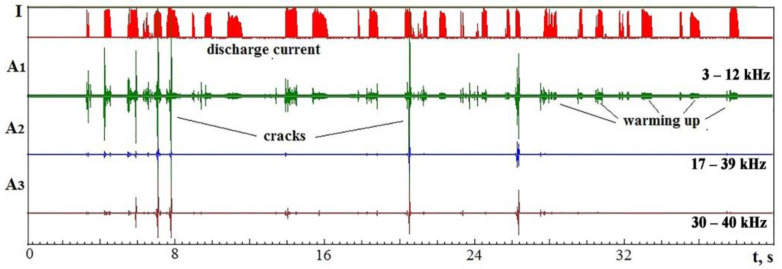
Records in time of the discharge current and VA signal components during electrical discharge die-sinking of IN23 ceramics.

**Figure 27 sensors-23-00750-f027:**
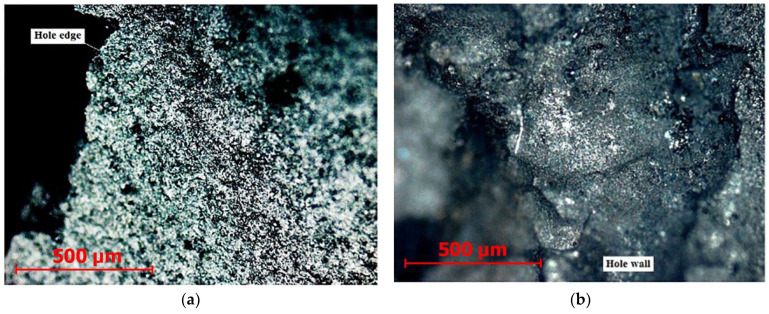
Photos of the hole surface when die-sinking a ceramic sample on an EDM machine: (**a**)—the edges of the hole with chips; (**b**)—hole walls with cracks.

**Table 1 sensors-23-00750-t001:** Parameters of VA signals for EDM of low-carbon steel.

Current, %	A_1_, mV	A_2_, mV	L, mm	*K_f_*
17	50.6	30.7	1.0	1.65
67	326	217	9.5	1.49
100	469	337	14	1.39

**Table 2 sensors-23-00750-t002:** Parameters of VA signals for EDM of IN23 ceramics.

Current, %	A_1_, mV	A_2_, mV	L, mm	*K_f_*
33	1.35	2.5	1.8	0.54
67	2.27	3.43	3.4	0.66
100	3.47	4.5	4.32	0.77

## Data Availability

Data sharing is not applicable to this article.

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
