# Peer review of "Investigation of the Information Possibilities of the Parameters of Vibroacoustic Signals Accompanying the Processing of Materials by Concentrated Energy Flows"

_sensors, 2023, doi:10.3390/s23020750_

Round 1

Reviewer 1 Report

The investigation in this paper has shown that vibration parameters can not only be used in monitoring systems to prevent negative phenomena during heat flow machining but also be a tool for understanding the kinetics of processes accompanying various heat processing technologies. So paper presents the very interesting and actual results in the point of engineering. I think, Sensors journal should publish this paper, as it's very interesting. 

But before that the following changes must be done by authors,

-> abstract is misleading. There is no clear sentence that will show what is the aim of this paper. What are the main results presented in quantitative way. So please revise. 

-> Literature review is done in incorrect way. Please do not cite multiple papers without deep investigation. The following issues must be revised (see bold) : "AE occurs during the plastic 75 deformation of solid materials [1-5], the development of defects in them [6-8], phase 76 transformations associated with a change in the crystal lattice [9-13], the formation of particles of a new phase in supersaturated solutions, when the boundaries of magnetic domains shift during crystallization and melting of a substance [14-17] and is widely used as a method of non-destructive testing in defectoscopy [18-22].

-> please add adequate references for the equations (expressions) used in this paper.

->Please discuss the presented results in the context of secondary literature. 

Author Response

Dear reviewer,

Thank you so much for your kind evaluation of our work. We agree with all your proposals and comments and have modified the manuscript accordingly. The editing was carried out throughout the text, corrections and additions were made, and the abstract and conclusions were written.

We hope the manuscript will be suitable for publishing in Sensors and attract many potential journal readers with your comments.

Kind regards,

Authors.

 Point 1: Abstract is misleading. There is no clear sentence that will show what is the aim of this paper. What are the main results presented in quantitative way. So please revise.

Response 1: Thank you for the fair remark regarding the abstract. The annotation size was reduced during editing to reduce the volume of the manuscript, and an important phrase was lost. The abstract has been revised. The authors hope that the current version looks appropriate.

Point 2: Literature review is done in incorrect way. Please do not cite multiple papers without deep investigation. The following issues must be revised (see bold) : "AE occurs during the plastic deformation of solid materials [1-5], the development of defects in them [6-8], phase transformations associated with a change in the crystal lattice [9-13], the formation of particles of a new phase in supersaturated solutions, when the boundaries of magnetic domains shift during crystallization and melting of a substance [14-17] and is widely used as a method of non-destructive testing in defectoscopy [18-22]."

Response 2: Thank you for noticing it. The authors agree with the reviewer that the cited literature is best subjected to in-depth research. However, this depends on the objectives of the publication. An in-depth study is necessary if an analytical review is done in the chosen direction. If research is being carried out to develop the ideas of other authors, then the essence of the previous publication must also be shown. If the conclusions of other publications in the article are refuted, then it is especially necessary to reveal the essence of the views of other authors. In this case, the references show that this statement is consistent with many publications and should not raise doubts in the reader. If only a reference is given, then this may not convince skeptics. With restrictions on the amount of text, we can either leave the references as they are or remove them.

Point 3: Please add adequate references for the equations (expressions) used in this paper.

Response 3: Thank you for the fair remark. Formula (1) is introduced to show the order of growth of maximum stresses with decreasing radius of the tool's cutting edge. The "~" sign denotes proportionality in this case. The expression was obtained from a theoretical calculation [31,32] for elliptical cracks experiencing a tensile load. The increase in maximum stresses with a decrease in the radius of the cutting edge is intuitively apparent. The formula is given only to show the approximate order of this growth. It is not intended for further use in calculations, especially when it is considered that the real cutting edge only approximately resembles a cylinder. References to works [31,32] have been added to the article. The formula was taken from [31], but it references the calculation method from [32], published in 1913. The authors of [31] mention the technique of C. E. Inglis [32].

  1. Ionov V.N., Selivanov V.V. Dynamics of destruction of a deformable body. - M.: Mashinostroenie, 1987. 272 pages.
  2. Inglis, K. E. “Stresses in Plates Due to Cracks and Sharp Corners,” Proceedings of the Institute of Naval Architects, Vol. 55, pp. 219-241, 1913.

Formula (2) was obtained through experiments, and there are references. The expression is given as an example and has a local character: if you take a different frequency range or install the accelerometer in a different place, it will change. It should be noted that the elastic system has many natural frequencies; their appearance depends on the accelerometer placement. However, the nature of the dependence remains. Here it was important to show the monotonous relationship between productivity and vibration amplitude.

Point 4: Please discuss the presented results in the context of secondary literature.

Response 4: Thank you for pointing it out. Changes have been made in the conclusions, emphasizing information from literary sources

Reviewer 2 Report

It is of great significance to analyze the changes of surface and substance material including structural and phase rearrangements, chemical reactions that cause local changes in the volume of substance, and abrupt changes in elastic stresses, during the processing of materials by concentrated energy flows. Monitoring of VA signal parameters is a tool for understanding the hard-to-observe kinetics of the processing of various materials by exposure to concentrated energy flows. The work has good theoretical significance and engineering application value. So i think this manuscript can be accepted. However, some problems can be improved in the manuscript. (1)It is better to reveal the relationship between vibroacoustic signals and material physical changes during processing with concentrated energy flows, such as laser machining, electrical discharge machining and electron beam machining. (2)Structural and phase rearrangements, chemical reactions and other physical changes effect the vibroacoustic signals. So it is important to analyze the quantitative and qualitative relations. (3)It would be better to add the comparison of vibroacoustic signals during in the laser machining, electrical discharge machining and electron beam machining, which have different physical changes. (4)It is better to be concise in the conclusion.

Author Response

Dear reviewer,

Thank you so much for your kind evaluation of our work. We agree with all your proposals and comments and have modified the manuscript accordingly. The editing was carried out throughout the text, corrections and additions were made, and the abstract and conclusions were written.

We hope the manuscript will be suitable for publishing in Sensors and attract many potential journal readers with your comments.

Kind regards,

Authors.

Point 1: It is better to reveal the relationship between vibroacoustic signals and material physical changes during processing with concentrated energy flows, such as laser machining, electrical discharge machining and electron beam machining.

Response 1: Thank you for the fair remark. The reviewer advises paying more attention to the relationship between vibration parameters and physical changes in the material. The authors agree that this is a critical issue, but its in-depth study requires labor-intensive research. So far, this problem has been solved at the first level: obtaining information about negative situations that accompany processing. Figure 2 shows the change in the vibration signal parameters with a change in the surface layer's deformation during the workpiece's turning. The article shows the identification of cracks that occur during processing (Figure 26) and the cooling of the surface layer (Figure 12), identification of situations preceding the wire tool breakage during electrical discharge machining (Figures 17, 23, 24). It should be noted that EDM fault detection (Figures 21, 22, 26) in ceramic machining can complement existing control techniques. Based on the conducted research, it is possible to analyze the processing modes of new materials for their correction to increase the processing productivity and quality of parts.

Point 2: Structural and phase rearrangements, chemical reactions and other physical changes effect the vibroacoustic signals. So it is important to analyze the quantitative and qualitative relations.

Response 2: Thank you for pointing it out. Figure 11 shows information about the relationship of vibration parameters with the useful output of the surface alloying. It is close to the reviewer's advice about analyzing quantitative and qualitative relationships. The figure shows the dependence of the vibration parameters on the fraction of the area covered with intermetallic compounds after irradiation in a vacuum chamber. It is a significant result for monitoring the doping process. If excess energy is applied, too much of the film that was previously deposed to the part may evaporate. If the power density turns out to be too low, the intermetallic compounds' formation reaction will not occur in the required amount. It also applies to other doping options in vacuum chambers, where there are no other sources of data and where the role of the random factor is significant. Preliminary experiments are required to determine the upper and lower limits of vibration signal parameter values.

Point 3: It would be better to add the comparison of vibroacoustic signals during in the laser machining, electrical discharge machining and electron beam machining, which have different physical changes.

Response 3: Thank you for the fair remark on the comparison of vibration signals for different types of processing with concentrated energy flows (reviewer's advice). It should be noted that it is difficult to compare the parameters of vibration signals for different technologies since different materials are being processed and different technological problems are being solved. It is despite all these technologies being based on the same processes associated with transferring energy of a high power density to matter. Within each technology, signal parameters are compared under different modes and conditions. For example, Figures 6 and 7 compare the vibration parameters accompanying laser melting processes (laser powder bed fusion) with different beam speeds. For all the technologies mentioned in the article, comparisons of vibration parameters are given for different power densities of thermal exposure. In Response to Point 1, it is noted in which situations the changes in the parameters of vibration signals are considered.

We agree with the reviewer's recommendation to expand knowledge about the relationship between the kinetics of the processes under consideration and the vibration parameters. However, our intentions do not always coincide with our capabilities. Therefore, we promise to continue this work.

Point 4: It is better to be concise in the conclusion.

Response 4: Thank you for the fair remark. The reviewer is right about the lengthy conclusion. It was shortened and revised.

Reviewer 3 Report

This paper attempts to use acoustic methods to detect device processing. There are complex physical and chemical processes when processing devices using concentrated energy. When using concentrated energy to process different materials, there will be different internal stress processes inside the material (related to the type of material and the mechanical structure of the device), which will lead to the vibration inside the device to generate acoustic signals. Non-destructive and efficient detection methods have always been an important part of the research field. Under the premise of clear constraints, it is of great significance to invert the device processing information through acoustic signals. For this paper:

(1)   The main method of acoustic signal processing in this paper is the Fourier transform method. This method can efficiently extract the frequency domain signal, but completely abandon the time domain information. The machining process is usually the process of applying different energies to the device according to a predetermined procedure in a chronological order. It may occur that some frequency signals are abnormal signals at a certain stage and are not abnormal signals at a certain stage. Fourier transform method can only analyze whether there is a frequency signal in a certain time period, but can not obtain the distribution of the frequency signal in the time period. It is expected that the author can further explain the applicable scope of the method proposed in this paper and the constraints of the analysis method.

(2)   The authors have given several formulas similar to empirical formulas in the paper. It is expected that the authors can further explain the fitting process of these formulas in detail.

(3)   It is expected that the author can analyze the experimental results in terms of statistics and obtain the information of confidence.

(4)   The mathematical formulas in this paper should be checked. The punctuation at the end of the formula should be noted.

(5)   For the Introduction and literature review section, the author is requested to carefully verify the degree of correlation between some of the relevant references and the research content of this paper.

The conclusion of the reviewers is that the current version of the paper is not acceptable, and the authors are expected to resubmit the paper after an MAJOR REVISION.

Author Response

Dear reviewer,

Thank you so much for your kind evaluation of our work. We agree with all your proposals and comments and have modified the manuscript accordingly. The editing was carried out throughout the text, corrections and additions were made, and the abstract and conclusions were written.

We hope the manuscript will be suitable for publishing in Sensors and attract many potential journal readers with your comments.

Kind regards,

Authors.

 Point 1: The main method of acoustic signal processing in this paper is the Fourier transform method. This method can efficiently extract the frequency domain signal, but completely abandon the time domain information. The machining process is usually the process of applying different energies to the device according to a predetermined procedure in a chronological order. It may occur that some frequency signals are abnormal signals at a certain stage and are not abnormal signals at a certain stage. Fourier transform method can only analyze whether there is a frequency signal in a certain time period, but can not obtain the distribution of the frequency signal in the time period. It is expected that the author can further explain the applicable scope of the method proposed in this paper and the constraints of the analysis method.

Response 1: Thank you for pointing it out. The reviewer's remark is correct that spectral analysis is insufficient to observe jump-like processes under impulsive action on the matter. The article shows not only the spectra of vibration signals but also records of changes in vibration parameters over time (Figures 12, 16, 17, 21-26). The spectra were used to select the most suitable frequency ranges for subsequent temporal analysis.

It is crucial to observe changes in amplitude over time when volumes, internal stresses, and other characteristics change abruptly. However, it is necessary first to use the signal spectrum analysis to select the most informative frequency ranges. These ranges are chosen based on experiments. The signal-to-noise ratio in these ranges should be satisfactory, and the amplitudes should be sensitive to the changes in the conditions of the processing area. Choosing such frequency ranges in which the amplitude changes will not repeat each other, differing only in scale, is important. Therefore, one range was chosen in relatively low frequencies, and the second was in the region of high frequencies.

A paragraph that explains it was added to the text of the manuscript. Explanations were added to the figures with signal records and their parameters in the captions throughout the manuscript's text. The abstract and conclusions were rewritten.

 Point 2: The authors have given several formulas similar to empirical formulas in the paper. It is expected that the authors can further explain the fitting process of these formulas in detail.

Response 2: Formula (1) is introduced to show the order of growth of maximum stresses with decreasing radius of the tool's cutting edge. The "~" sign denotes proportionality in this case. The expression was obtained from a theoretical calculation [31,32] for elliptical cracks experiencing a tensile load. The increase in maximum stresses with decreasing edge radius is intuitively apparent. The formula is given only to show the approximate order of this growth. It is not intended for further use in calculations, especially when it is considered that the real cutting edge only approximately resembles a cylinder. References to works [31,32] have been added to the article. The formula is taken from [31], but it references the calculation method from [32], published in 1913. The authors of [31] mention the technique of C. E. Inglis [32].

  1. Ionov V.N., Selivanov V.V. Dynamics of destruction of a deformable body. - M.: Mashinostroenie, 1987. 272 pages.
  2. Inglis, K. E. “Stresses in Plates Due to Cracks and Sharp Corners,” Proceedings of the Institute of Naval Architects, Vol. 55, pp. 219-241, 1913.

Point 3: It is expected that the author can analyze the experimental results in terms of statistics and obtain the information of confidence.

Response 3: Thank you for the remark. Formula (1) is discussed in Point 2. Formula (2) was obtained experimentally; its confidence interval can only be discussed concerning the conditions of the experiment. The method for determining the volumes of removed material had the most significant error. The uncertainty of the technique in the form of a 95% confidence interval was estimated as 8-11% of the mathematical expectation. It should be noted that formula (2) is an example of a monotonic increase in the amplitude of vibrations with an increase in productivity during laser processing. In the article, it is given for the RMS amplitude in the 16 kHz octave. If a different frequency range is taken or the sensor is placed on another machine, the conditions will change, and the dependence form will differ, but the monotonous character of growth will remain since it has a principal character. It depends mainly on natural frequencies, which are different for different elastic systems and even for different accelerometer positions. For practical application, the position of the accelerometer should be constant.

 Point 4: The mathematical formulas in this paper should be checked. The punctuation at the end of the formula should be noted.

Response 4: Thank you for noticing it. In formula (1), a typo that appeared during editing was corrected. Formula (2) was obtained using the simplest approximation of points obtained at different laser powers. The power dependence was chosen as the most illustrative; the approximation was carried out using the least squares method. It should be noted that formulas (1) and (2) show the nature of dependencies.

Point 5: For the Introduction and literature review section, the author is requested to carefully verify the degree of correlation between some of the relevant references and the research content of this paper.

Response 5: Regarding the list of references, few works are devoted to studying the relationship between vibration parameters and the characteristics of the processing using concentrated energy flows. Concerning electron-beam doping, they do not exist at all. For many readers, the presence of vibration signals in EDM or electron beam processing is new. Therefore, references were made to works where there were experiments confirming the statements made in the article, although the works themselves pursued other goals. Links to the works mentioned in paragraph 2 have been added to the list.

The manuscript was checked by a native English-speaking colleague and networked by https://www.grammarly.com/
The report (grammarly_report.pdf) is attached.

Round 2

Reviewer 1 Report

Paper can be accepted now.

Reviewer 3 Report

The authors have solved all the comments.